

# Factors affecting the relative abundance in an overfished stock: red grouper (*Epinephelus morio*) in the Southeastern Gulf of Mexico

Iván Oribe-Pérez[1], Iván Velázquez-Abunader[1] and Carmen Monroy-García[2]

[1] Departamento de Recursos del Mar, Centro de Investigación y de Estudios Avanzados del Instituto Politécnico Nacional, Mérida, Yucatán, Mexico
[2] Centro Regional de Investigación Acuícola y Pesquera de Yucalpetén, Instituto Nacional de Pesca, Yucalpetén, Yucatán, Mexico

Corresponding author
Iván Velázquez-Abunader,
jvelazquez@cinvestav.mx

## ABSTRACT

The most important fisheries are recording catches below their historical averages despite increased effort. This level of overfishing is worrying and requires the establishment of feasible and precise measures to prevent a continuing decrease in biomass. Determining the factors that lead to changes in the abundance and distribution of overfished resources would allow us to identify the strengths and weaknesses of management schemes; this approach would also make it possible to estimate more accurate parameters for their evaluation. We hypothesize that environmental, temporal, spatial, and operational components contribute to the variation in the relative abundance. Thus, we analyzed the red grouper fishery, the most important demersal fishery in the southeastern Gulf of Mexico (SGM); it is locally known as *escama*. We employed the catch per unit effort (CPUE) as an index of relative abundance recorded by the semi-industrial fleet (kilogram per effective fishing day) and the small-scale fleet (kilogram per effective fishing hour) during the overexploitation phase (from 1996 to 2019). We fitted several variables of the components using generalized additive models (GAM) and used multi-model inference to determine the best GAM for each fleet. For both fleets, the operational and temporal components (fishing gear and year) have had a greater impact on the distribution and abundance of red grouper in the SGM than the spatial and environmental components (the place of origin and sea surface temperature). These findings encourage the exploration of métier schemes for more efficient fishery management. In addition, we have identified several strategies that would support the recovery of the resource, such as restricting fishing in the quadrants located to the northeast or regulating scuba diving. We recommend that in the future, researchers use the indices we have generated in the present study to evaluate the red grouper fishery.

## INTRODUCTION

Toward the end of the last century, about one third of the main fishery resources recorded declines in catches, with low levels of biomass and catches, attributed to the increase in fishing effort and the modernization of fishing fleets in the 1970s and 1980s (*Grainger & Garcia, 1996*; *Pauly et al., 2002*). Overfishing of several fish stocks is a growing concern to the fishing industry and decision-makers (*FAO, 2020*). The level of overexploitation that commercial fisheries face requires management that ensures the responsible use of living marine resources based on stock assessments and supervision of resource dynamics (*Hilborn & Walters, 1992*; *Forrestal et al., 2019*). This approach should not neglect governance and should recognize the importance of ecological or environmental factors, including the effects of climate change (an increase in the ocean surface temperature, sea level rise, and ocean acidification, among others) and its implications on primary productivity, as well as the socioeconomic influences inherent to fishing activity. For this reason, it is necessary to know and understand what factors can produce spatial and temporal changes in the relative abundance of resources. Globally, the relative abundance or catch per unit effort (CPUE) is commonly assumed to be proportional to abundance in fish stock assessments ($\beta \approx 1$) (*Hilborn & Walters, 1992*; *Maunder & Punt, 2004*). An increase or decrease in the CPUE may reflect changes in resource abundance and biomass (*Forrestal et al., 2019*). However, it is common for this assumption not to hold because there are states of hyperdepletion or hyperstability, which makes the CPUE an unreliable indicator of the abundance of the stock (*Maunder & Punt, 2004*).

CPUE is a key component of stock assessment models (*Hilborn & Walters, 1992*; *Forrestal et al., 2019*). In this regard, different authors have pointed out that, to strengthen the assessments and management of exploited resources, it is important to use an index of abundance that is fishery independent or to standardize nominal CPUE values; however, fishery-independent data are often extremely expensive or difficult to collect (*Maunder & Punt, 2004*; *Hua et al., 2019*). Fisheries management decisions must be made in real time and with the best scientific evidence. In this context, nominal CPUE values may provide limited information on the state of the resource (*Quinn & Deriso, 1999*; *Maunder & Punt, 2004*; *Haddon, 2011*).

To generate more reliable and representative data for stock assessments, nominal CPUE values should be standardized (*Quinn & Deriso, 1999*; *Maunder & Punt, 2004*; *Hua et al., 2019*). This standardization should consider environmental, temporal, spatial, and operational effects on the catchability coefficient so that the CPUE is a reliable indicator of the abundance of the stock (*Watters & Deriso, 2000*; *Maunder & Punt, 2004*; *Hua et al., 2019*). In the last few decades, there has been much effort to solve the problems associated with CPUE fitting. Several statistical models have been considered, including generalized linear models (GLM); generalized additive models (GAM); generalized linear mixed models (GLMM); and generalized additive models of location, scale, and shape (GAMLSS) (*Watters & Deriso, 2000*; *Maunder & Punt, 2004*). These statistical models are also often used in ecology to explore, analyze, and understand the complexities and interactions of covariates, thus facilitating more accurate results and the assessment of data significance

(*Zuur & Ieno, 2016*). In the last several years, the use of machine learning methods has led to significant improvements in model construction and fitting for CPUE standardization (*Yang et al., 2020*). These methods have proved to be efficient to determine the behavior of the CPUE with respect to different variables (*Maunder & Punt, 2004*; *Tian et al., 2009*; *Hua et al., 2019*).

## The importance and vulnerability of groupers: the ecological role of red grouper

Groupers are highly valued in the international market for the quality of their meat; hence, they have a high commercial value (*Sadovy de Mitcheson et al., 2013*, *2020*; *Mavruk et al., 2018*). Groupers are also of great importance for the livelihood and food security in coastal communities. They play a crucial ecological role in reef ecosystems, as they are top predators and maintain balance in the food chain (*Sadovy de Mitcheson et al., 2013*, *2020*; *Mavruk et al., 2018*). However, they are also among the species most vulnerable to fishing pressure due to the characteristics of their lifecycle, such as a slow growth rate, late sexual maturation, protogynous hermaphroditism (sex change from female to male), and great longevity (*Sadovy de Mitcheson et al., 2013*, *2020*; *Mavruk et al., 2018*).

In the SGM, red grouper (*Epinephelus morio*) is known as an "ecosystem engineer" because its behavior affects habitat dynamics and ecosystem resilience (*Coleman & Williams, 2002*; *Coleman et al., 2010*). Through its excavations, this species creates holes that serve as crucial spaces for various species and some of commercial interest such as the Caribbean spiny lobster (*Panulirus argus*) (*Coleman et al., 2010*). These holes also serve as cleaning stations, where juvenile blue angelfish (*Holacanthus bermudensis*) and queen angelfish (*Holacanthus ciliarus*) remove external parasites from other species. Besides, the red grouper provides protection from roaming predators to the species cohabiting with them in their holes (*Coleman et al., 2010*). In addition, the burrowing behavior of the red grouper influences sediment biogeochemistry and organic matter decomposition (*Coleman & Williams, 2002*). Given the multifaceted role of the red grouper in the ecosystem as a top predator in the food chain and as an ecosystem engineer, a decline in these populations (in biomass and number) would compromise its existence and lead to direct and indirect impacts on the biodiversity of the ecosystem, including effects on water and sediment processes (*Coleman & Williams, 2002*).

## Fishery background

The red grouper has traditionally been the main target species of a multispecies demersal fishery in the SGM. This highly valued species in the national and international market has sustained an established fishery in the Yucatan Peninsula (*Diario Oficial de la Federación (DOF), 2018*; note that DOF is the Spanish acronym for the Official Journal of the Federation, the official organ of the Mexican government for the publication of laws, decrees, regulations, agreements, and other provisions of the branches of the State). The historical trends of red grouper catches have shown the general development phases of a fishery: growth, exploitation, and overexploitation (Fig. 1) (*Hilborn & Walters, 1992*). The growth phase (1958–1978), which recorded the highest historical production (~20,000

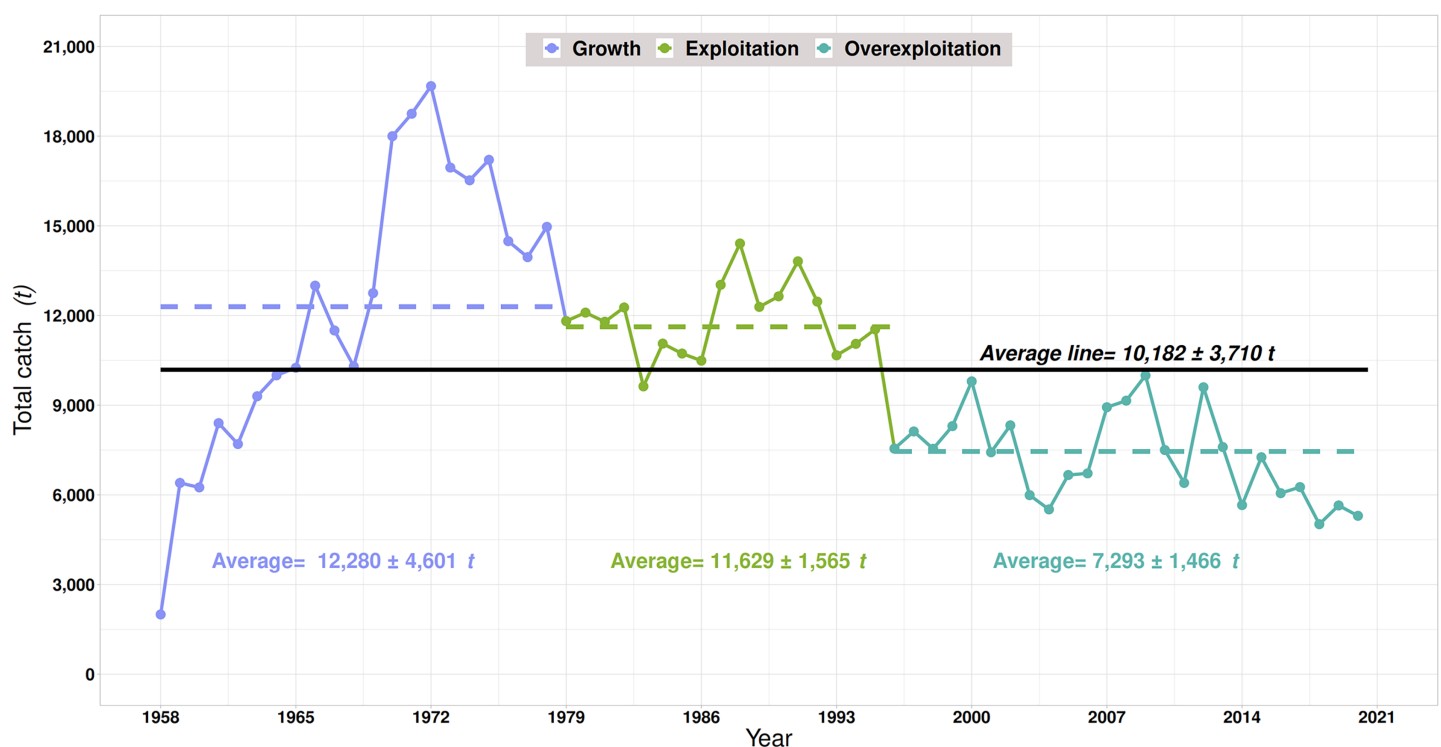

**Figure 1** Behavior of historical catches of the red grouper *Epinephelus morio* in the Campeche Bank, Mexico.

t), was characterized by investment, technological development, and the entry of new vessels (*Monroy-García, Galindo-Cortes & Hernández-Flores, 2014*). The exploitation phase (1979–1995), with an evident decreasing trend in catches, showed high fishing pressure, mainly on juveniles (*Arreguín-Sánchez, 1987*; *Diario Oficial de la Federación (DOF), 2014*). During the overexploitation phase (1996–2020), the lowest catch levels in this fishery's more than six decades of development were recorded. During this period, the red grouper was reported to be overfished in the GSM (*Monroy-García, Galindo-Cortes & Hernández-Flores, 2014*), and according to the International Union for Conservation of Nature (IUCN) Red List, this already endangered species has been recategorized to vulnerable (*Brule et al., 2018*; *Sadovy de Mitcheson et al., 2020*).

The red grouper fishery in the SGM involves four fleets with different fishing power. The fleets catch different components of the stock (juveniles and adults) due to the size segregation of the red grouper with depth and the area in which the fleets operate. There are two Mexican commercial fleets as well as a small-scale fleet of approximately 4,200 boats, with lengths ranging from 6 to 7.5 m, and outboard engine power from 40 to 115 HP. This fleet targets primarily juvenile fish because it operates below the 40 m isobath. Another semi-industrial fleet composed of 456 vessels with *escama* fishing licenses (which are used to catch red grouper and various species of commercially valuable demersal fish in the region). The length of the semi-industrial vessels is 10–23 m, and the engine power is

75–350 HP. The semi-industrial fleet focuses its efforts on adults due to their autonomy and spatial distribution, which extends from a dept of 40–200 m. The Cuban fleet (22 m in length) is also semi-industrial (*Monroy-García, Galindo-Cortes & Hernández-Flores, 2014*). It lost its fishing licenses in the Campeche Bank in 2022 because of the lack of red grouper surplus (additional amount of resource available for exploitation) as established by the LGPAS (Spanish acronym, General Law of Sustainable Aquaculture and Fisheries; *Diario Oficial de la Federación (DOF), 2007*). The LGPAS is a legal instrument of the Mexican government to promote and manage the exploitation of fishery and aquaculture resources in its national territory. There is also a Mexican sport-recreational fishing (*López-Rocha, Vidal-Hernández & Bravo-Calderón, 2020*), whose the number of fishers, the vessel sizes and catch levels are unknown.

In this study, we tested the hypothesis that different environmental, temporal, spatial, and operational factors affect the CPUE for the red grouper stock in the southeastern Gulf of Mexico (SGM). Therefore, we aimed to define what factors contribute to the variability of relative abundance in overfished fisheries, such as the red grouper fishery in the SGM, to understand the dynamics of the resource and to identify key variables that can improve its management schemes, focusing on a possible recovery of a fishery that is in critical condition.

## MATERIALS AND METHODS

### Study area

The Campeche Bank is located in the SGM (20–24°N, 86–93°W) with an area of 175,000 km$^2$ and is bounded by the 200 m isobath and coastline. Its substrate is characterized by limestone sediments, muddy bottoms, sands, and shells, accompanied by irregular configurations formed by cays and coral reefs (*Monroy-García et al., 2019*; *López-Rocha, Vidal-Hernández & Bravo-Calderón, 2020*). The zoning established by INAPESCA (Spanish acronym, National Fisheries Institute) was used to classify the operating range of the semi-industrial fleet. As the Mexican fisheries authority, INAPESCA is responsible for conducting scientific studies on fisheries, including the red grouper fishery. For this purpose, it has divided the SGM into 20 fishing zones (Fig. 2). The small-scale fleet was classified according to the place of origin of the boats (Fig. 2) to identify fishing areas and the distribution of fishing effort in the red grouper fishery in the SGM.

### Data records

Data were obtained from two sources, both of which were compiled and provided by INAPESCA (inconsistent and incomplete data were not used). The first data source for the semi-industrial fleet is records kept by vessel skippers in the fishing logbooks from 1996 to 2019 (2011 had no records, because the information was not available in the data source to which we had access). The fishing logbooks are mandatory registration and control documents that skippers must submit to the competent authorities to maintain and renew their fishing licenses (each fishing logbook represents a fishing trip). In each fishing logbook, the skipper records fishing activities, including the name of the ship and skipper,

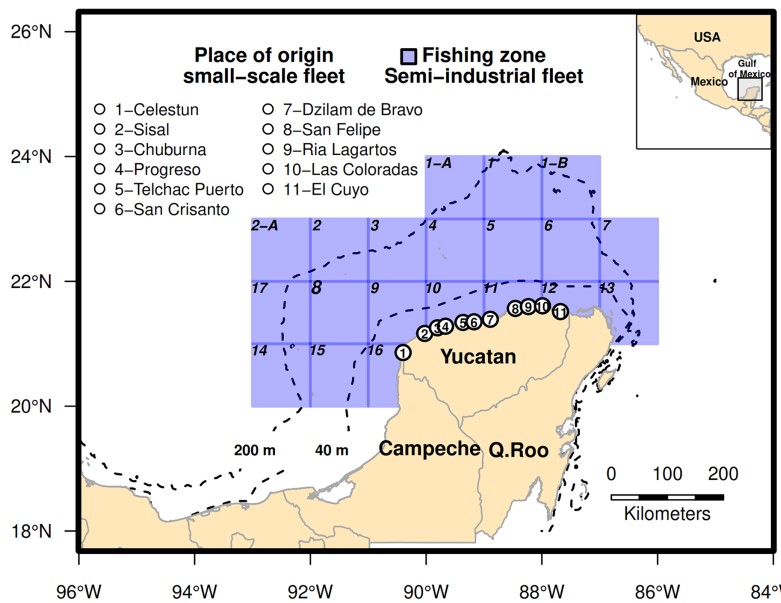

**Figure 2 Fishing area for the catch of red grouper in the southeastern Gulf of Mexico.** The spatial component for the small-scale fleet was addressed by the place of origin of the boats, and in the semi-industrial fleet, the fishing zones proposed by the Instituto Nacional de Pesca were used. Dotted lines indicate the 40 m isobath and 200 m common operating range of the small-scale and semi-industrial fleets, respectively.

catches by species (kg), the depth and fishing quadrant, fishing gear, the date of departure and arrival, the effective days of fishing, and the crew size. In this study, on average the trips of 375 vessels of the semi-industrial fleet were analyzed per year, which is equivalent to 82% of the vessels licensed for *escama* fishing, recording 46,542 fishing trips, which represents 78% of the total number of trips according to the SIPESCA platform (Spanish acronym, System of Information on Fisheries and Aquaculture). SIPESCA is the Mexican government's official platform for the control and registration of fishing and aquaculture activities (https://sipesca.conapesca.gob.mx/loginFIEL.php).

The second data source corresponds to random monthly interviews of fishers who are part of the small-scale fleet; they were conducted in the main landing ports of the Yucatan coast from 2013 to 2019 (Fig. 2). The interviews were structured to allow for a systematic application—that is, all interviewees were asked the same set of closed questions (*Bernard, 2017*). The fishers were interviewed upon arrival at the port to obtain general information on the fishing operation in relation to the target species; the total catch (kg) and catch by species, the fishing zone (course, distance to port, and depth), fishing gear, engine power, crew size, fishing hours, and fishing costs were also recorded. Each landing port was visited for 3–4 days per month. The sampling approach adopted in that study was opportunistic (*Salas, Torres-Irineo & Coronado, 2019*), due to various circumstances that affect the number of trips taken by the small-scale fleet per week and month, including the decisions of individual fishers to go or not go fishing. These decisions are influenced by market demand, profits obtained in previous days, opportunity costs, and weather conditions
(*Salas, Sumaila & Pitcher, 2004*; *Salas, Torres-Irineo & Coronado, 2019*). Other relevant factors—resource regulations in the SGM (for example, during octopus or lobster season, the effort to catch red grouper decreases), the availability of bait, and temporary closures of ports ordered by authorities due to adverse environmental conditions—restrict the navigation days of the small-scale fleet. Because of the above factors, the information collected varied by landing port. This study considered 2,775 interviews conducted with fishers from the small-scale fleet.

The semi-industrial fleet carries out fishing trips lasting 15–26 days and uses at least four different types of fishing gear: dinghy and short longline (DS), longline (LL), bicycle (B), and other vessels that use combined fishing gear methods (CFG). DS is used on mother ships carrying dinghies (wooden boats 3 m in length, locally named *alijos*, without motors and operated by a fisher). Each dinghy is released and operated independently in the fishing area with a short longline encompassing 80–100 hooks. Another vessel in the fleet uses LL with 500–3,000 hooks; it consists of a hydraulically driven reel with the mainline that operates directly from the vessel. Another is the B gear, which is a manual mechanism similar to a bicycle that operates with a mainline that has 4–6 hooks (*Monroy-García, Salas & Bello-Pineda, 2010*; *Quijano et al., 2018*). It has been reported that 90% of the total catch of the semi-industrial fleet is landed in the ports of Progreso and Yucalpeten, Yucatán (*Diario Oficial de la Federación (DOF), 2014*).

The small-scale fleet makes round trips with an average duration of 8 ± 2 effective fishing hours and uses different fishing gear, such as short longline (SL) composed of a main line from 300 to 3,000 m with 100–1,000 hooks; they are generally bottom longlines, so the extremes are placed with weights or sinkers. Scuba diving (SD) targets Caribbean red lobster (*Panulirus argus*) but also catches a high percentage of *escama* such as red grouper, hogfish (*Lachnolaimus maximus*), and black grouper (*Mycteroperca bonaci*). Divers mainly use a low-pressure compressor, which consists of a hose approximately 100 m long and a regulator that supplies air to the diver (hookah system). Handlines (H) are generally composed of a line of <100 m with 1–3 straight hooks of different sizes accompanied by a sinker; they are occasionally employed when gillnets (GH) or jimba (JH) are used. Jimbas are 2–8 m bamboo poles that usually have up to five lines with crabs as bait (used during the octopus season) (*Avendaño et al., 2019*). Gillnets are rectangular in shape and up to 100 m long with floats at the top, sinkers at the bottom, and buoys at the extremes (aimed at catching sardines, snook, or croakers).

## Environmental data

The sea surface temperature (average and standard deviation) data were obtained as monthly composites from the International Comprehensive Ocean-Atmosphere Data Set (ICOADS) at $1 \times 1°$ (*Freeman et al., 2019*). These compositions are consistent with the size of the fishing zones proposed by INAPESCA. The files were downloaded from https://icoads.noaa.gov/ in HDF format and displayed with the *raster* package (*Hijmans, 2020*) in R (*R Core Team, 2023*). The ICOADS data are *in situ* observations from ships, buoys, research reports, and other platforms (*Freeman et al., 2019*).

**Table 1 Classification of variables into components for modeling the catch per unit effort of red grouper in the SGM.**

| Component | Variable | Domain | Importance |
|---|---|---|---|
| Environmental | Sea surface temperature | Celsius degrees (°C) | Most models used to assess fish populations have inferred that changes in abundance are due to fishing pressure, ignoring a natural decline due to environmental changes (*Hilborn & Mangel, 1997*). Identifying the environmental factors that contribute to CPUE variation is key to incorporating this component into evaluation models (*Bigelow, Boggs & He, 1999*) |
| | Deep | Meters (m) | |
| Temporal | Year | Semi-industrial fleet :1996–2019 Small-scale fleet:2013–2019 | The objective of CPUE standardization is to obtain an accurate annual index for use in stock assessment models (*Hinton & Maunder, 2004*). Fishermen in the SGM modify their effort according to the fishing season and market conditions (*Salas, Torres-Irineo & Coronado, 2019*) |
| | Month | Semi-industrial fleet: January–December Small-scale fleet: February–December | |
| Spatial | Zone | Semi-industrial fleet :1–17 | Catchability varies spatially due to changes in fishery composition and resource abundance (*Maunder & Punt, 2004*; *Monroy-García et al., 2019*) so identifying where fishing occurs and the concentration of effort is key to regulating the fishery (*e.g.* no-take zones) |
| | Place of origin | Small-scale fleet: Celestun–El Cuyo | |
| Operative | Navigation days | Semi-industrial fleet: Days | Vessel skippers and fishermen modify their fishing trip based on their experience with the objective of maximizing their income, alternating fishing gear, changing target species, reducing fishing time and the crew size (*Murillo-Posada, Salas & Velázquez-Abunader, 2019*; *Salas, Torres-Irineo & Coronado, 2019*) |
| | Crew size | Semi-industrial fleet: Numbers | |
| | Fishing gear | Semi-industrial fleet: Bicycle (B), longline (LL), Dinghy and shortline (DS), Combined (C) Small- Scale fleet: Short longline (SL), Headlines (H), Scuba diving (SD), Jimba and handlines (JH), Gillnets and handlines (GH) | |

## Variables used for modeling

Four components were considered, and variables that could have an effect on the spatial and temporal variation of the red grouper CPUE in the SGM were selected (Table 1). A multicollinearity test using the generalized variance inflation factor (GVIF) method included in the *car* package (*Fox & Weisberg, 2011*) in R was applied to ensure that the candidate variables were not correlated. A GVIF ≥ 3 was considered to indicate collinearity problems (*Franke, 2010*; *Gareth et al., 2013*). Candidate variables to describe CPUE variation did not present collinearity (GVIF < 3), and all variables were considered in the models.

## Relative abundance index modeling

The indexes of relative abundance based on the CPUE were established to cover the diversity of fishing gears used by the fleets catching red grouper in the SGM. This approach aimed to achieve a comprehensive understanding of the dynamics of each fishing fleet and to establish a solid basis for their management. For the semi-industrial fleet, the CPUE was defined as kilograms of red grouper caught per effective fishing day (kg $EFD^{-1}$). For the small-scale fleet, the CPUE was defined as kilograms of red grouper per effective fishing hour (kg $EFH^{-1}$). Generalized additive models (GAMs) were used to model the dependent variables (CPUE) because they showed better fit and explanatory power (*Tian et al., 2009*; *Hua et al., 2019*). Due to there being no zero-catch data, it was assumed that the CPUE

data could fit some exponential family distribution. The Cullen and Frey plot representing the kurtosis and skewness of the data was used and non-parametric bootstrapping was performed to consider the uncertainty of the estimated values using the *fitdistrplus* package (*Delignette-Muller et al., 2015*) of the R programming language (*R Core Team, 2023*).

In the semi-industrial fleet, the CPUE did not fit the distributions (Fig. S1A). Thus, its distribution was assumed to be log-normal (Fig. S1B), and the identity link function was applied, $\mu = \eta = \ln (\text{CPUE})$. The residual plot of the model confirmed that the Gaussian error distribution selection was correct (Fig. S1C). Conversely, the CPUE of the small-scale fleet was fitted to the gamma distribution, and the inverse link was applied, $\mu = \dfrac{1}{\eta} = \dfrac{1}{\text{CPUE}}$ (Fig. S2).

The GAMs were built with the *mgcv* package (*Wood, 2021*) of R (*R Core Team, 2023*). Continuous variables were fitted with spline functions (*s*; sea surface temperature and depth), and categorical variables were assumed to have a linear fit (year, month, zone, place of origin, navigation days, crew size, and fishing gear). The purpose was to select the explanatory variables of the best candidate model, considering all possible model combinations. For the above, the *dredge* function (from the *MuMIn* package in R; *Bartoń, 2022*) that uses the multi-model approach based on the information theory (*Burnham & Anderson, 2002*) was used. This approach ranks the most parsimonious models according to the lowest value of the Akaike information criterion (AIC, *Akaike, 1983*), and calculates Akaike's differences ($\Delta_i$) and Akaike's weights ($w_i$). For each model, the deviance explained (D%), the pseudo coefficient of determination (Pseudo-$R^2$), and the adjusted pseudo coefficient of determination (adjusted Pseudo-$R^2$) were calculated.

For each fleet, the median nominal CPUE (original CPUE) was compared with the median fitted CPUE of the best GAM (the one with the lowest AIC) by year and month by using Wilcoxon's test (*Sokal & Rolf, 1981*).

## RESULTS

The most used fishing gear by the semi-industrial vessels was LL (73.06%), followed by DS (15.64%), B (9.85%), and CFG (1.45%). The small-scale fleet order was SD (54.20%), H (24.93%), SL (18.49%), JH (1.59%), and GH (0.79%). In the semi-industrial fleet, the median red grouper CPUE was 87.86 kg $\text{EFD}^{-1}$ (0.10–2548 kg $\text{EFD}^{-1}$). In the small-scale fleet, the median red grouper CPUE was 1.15 kg $\text{EFH}^{-1}$ (0.04–20 kg $\text{EFH}^{-1}$).

We fitted a total of 256 candidate models for the semi-industrial fleet and 128 candidate models for the small-scale fleet. Multi-model inference indicated that there is only one statistically viable model to explain variations in the red grouper CPUE for each fleet ($w_i >$ 0.90, Tables S1 and S2). For the semi-industrial fleet, the best GAM included year, month, zone, fishing gear, navigation days, *s*(depth), *s*(sea surface temperature), and crew size as explanatory variables, with D% = 39.33, Pseudo-$R^2$ = 0.39, and adjusted Pseudo-$R^2$ = 0.43. For the small-scale fleet, the best GAM included year, month, place of origin, fishing gear, crew size, *s*(depth), and *s*(sea surface temperature), with D% = 31.93, Pseudo-$R^2$ = 0.38, and adjusted Pseudo-$R^2$ = 0.40. The best models for the red grouper CPUE are summarized in Tables 2 and 3. The crew size in the small-scale fleet and the sea surface

**Table 2 Results of the best generalized additive model that explain the effects of the components on catch per unit of effort for red grouper caught by the semi-industrial fleet.**

| Component | Variable | df | edf | rdf | F |
|---|---|---|---|---|---|
| Environmental | s (Sea surface temperature) | | 8.46 | 8.91 | 2.84 |
| | s (Depth) | | 6.89 | 7.89 | 8.85 |
| Temporal | Year | 22 | | | 293.67 |
| | Month | 11 | | | 28.37 |
| Spatial | Zone | 18 | | | 14.30 |
| Operative | Navigation days | 31 | | | 31.50 |
| | Crew size | 18 | | | 118.70 |
| | Fishing gear | 3 | | | 2,557.93 |

Note:
Column heading abbreviations are as follows: degrees of freedom for parametric terms (df), effective degrees of freedom (edf), reference degrees of freedom (rdf) for smooth terms, statistics of the $F$ ($F$) model and $s$ denotes smooth terms.

**Table 3 Results of the best generalized additive model that explain the effects of the components on catch per unit of effort for red grouper caught by the small-scale fleet.**

| Component | Variable | df | edf | rdf | F |
|---|---|---|---|---|---|
| Environmental | s (Sea surface temperature) | | 6.56 | 7.68 | 2.41 |
| | s (Depth) | | 5.96 | 7.01 | 12.32 |
| Temporal | Year | 6 | | | 34.23 |
| | Month | 10 | | | 7.74 |
| Spatial | Place of origin | 10 | | | 3.43 |
| Operational | Crew size | 2 | | | 6.83 |
| | Fishing gear | 4 | | | 85.12 |

Note:
Column heading abbreviations are as follows: degrees of freedom for parametric terms (df), effective degrees of freedom (edf), reference degrees of freedom (rdf) for smooth terms, statistics of the F ($F$) model and $s$ denotes smooth terms.

temperature for both fleets were significant ($p < 0.05$), while all other variables were highly significant ($p < 0.001$). For both fleets, the variables with the largest effect size ($F$) were the fishing gear, followed by the year. The variables with the smallest effect size were the sea surface temperature and depth for the semi-industrial fleet and the sea surface temperature and place of origin for the small-scale fleet. This means that operational and temporal components have a greater impact on the red grouper CPUE than environmental and spatial components.

The best GAM for each fleet showed the influence of the explanatory variables in relation to the red grouper CPUE (Figs. 3 and 4). Regarding the environmental component, in the semi-industrial fleet, the relationship between the sea surface temperature and the red grouper CPUE was relatively stable between 24 and 30 °C. However, we observed inverse behavior of the sea surface temperature on the red grouper CPUE for each fleet. In the semi-industrial fleet, we noted high CPUE values at low sea surface temperatures (Fig. 3), while in the small-scale fleet, we found the highest CPUE values at the highest sea surface temperatures (Fig. 4). Depth showed a wide range in the red grouper CPUE, demonstrating that both fleets had moved into deeper waters over the
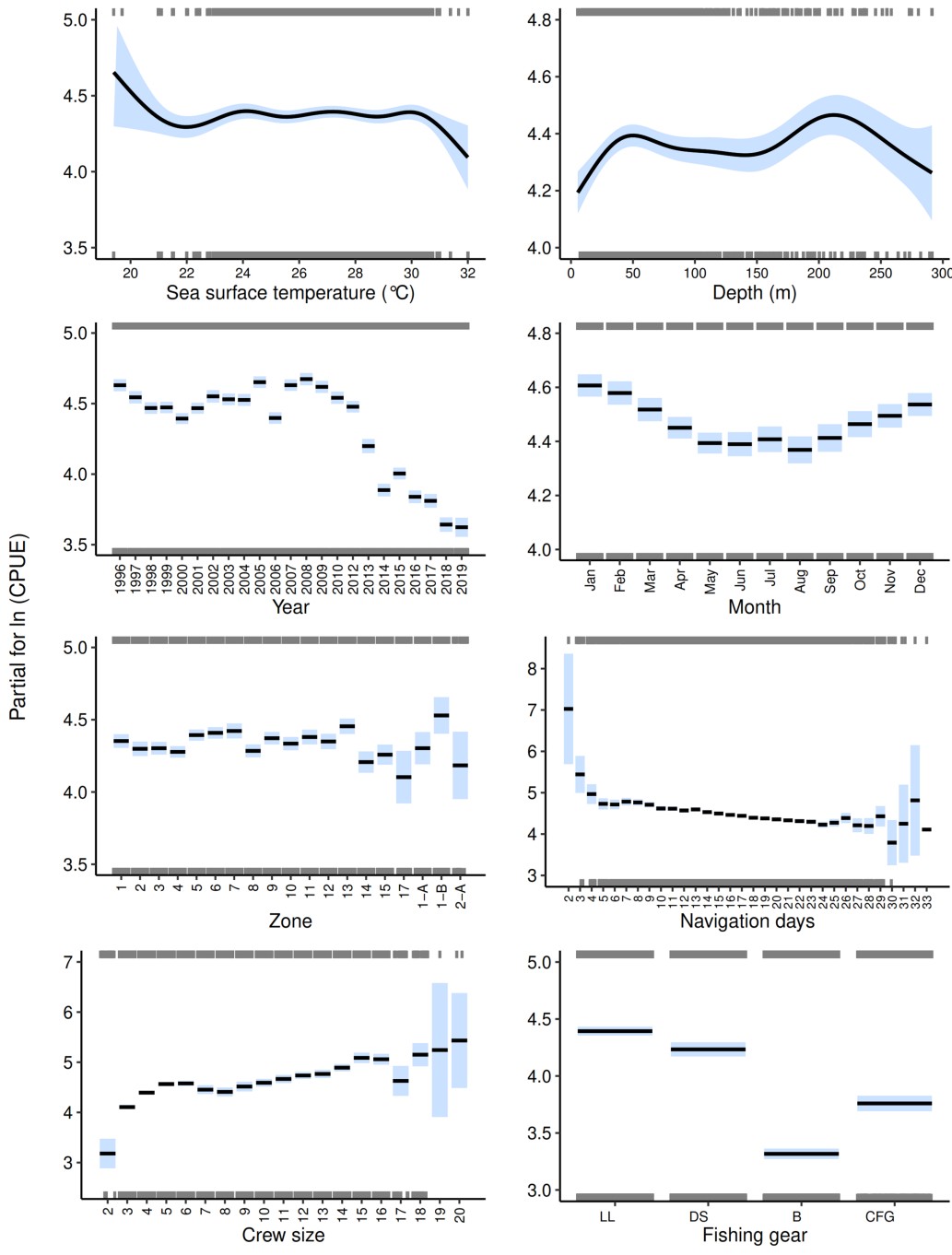

**Figure 3 Effects of environmental, temporal, spatial, and operational components on the catch per unit effort of red grouper recorded by the semi-industrial for the best GAM.** The solid black lines indicate the median catch per unit effort values, the blue shaded edges show the 95% confidence intervals, and the gray rugs on the upper and lower axes are the positive and negative residuals, respectively.

years due to a decrease in resources. However, going to deeper areas does not guarantee greater efficiency due to the notable uncertainty of the CPUE. Despite the overlap of the fleets when considering depths of less than 60 m, for the small-scale fleet, there were high

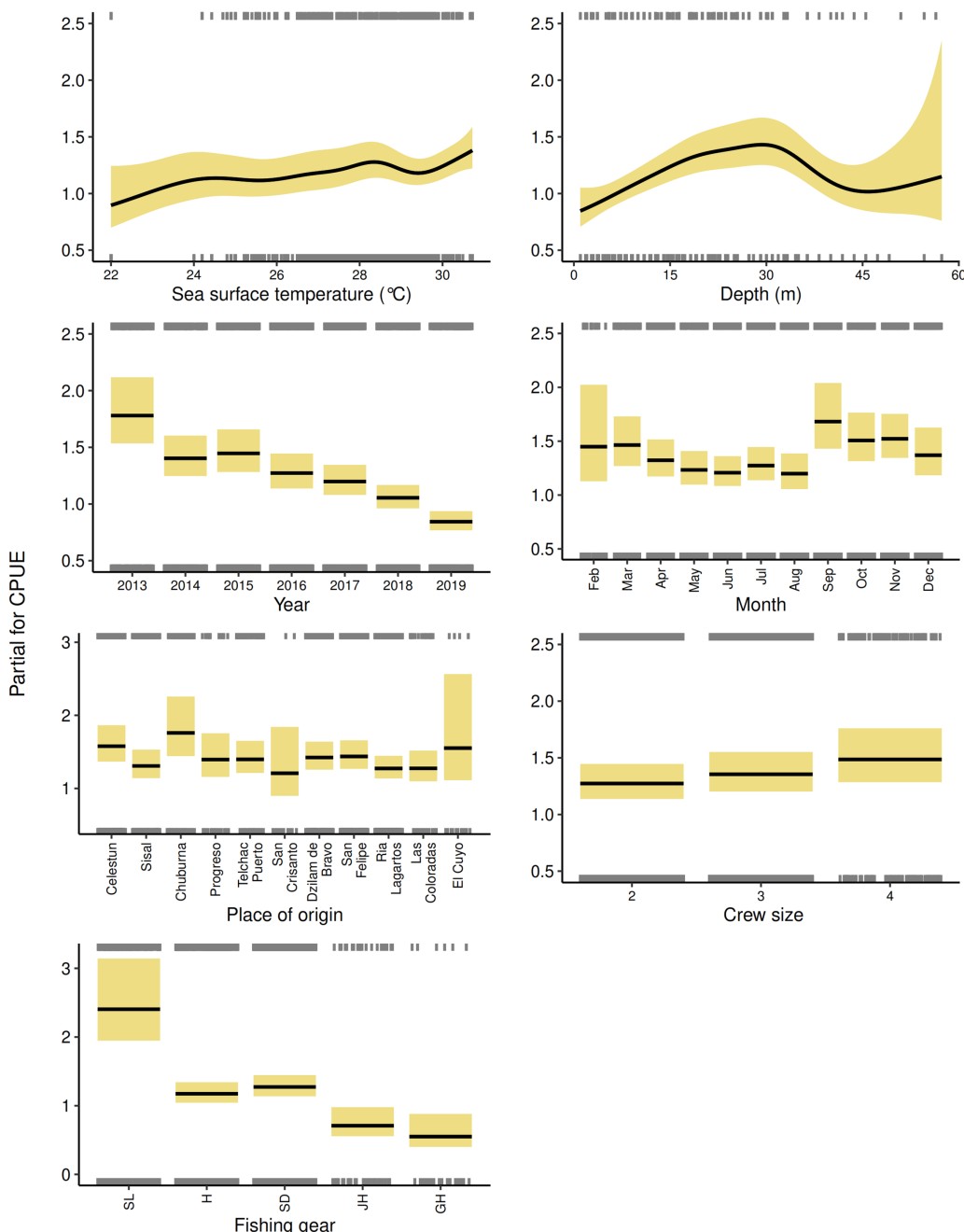

**Figure 4  Effects of environmental, temporal, spatial, and operational components on the catch per unit effort of red grouper recorded by the small-scale fleet for the best GAM.** The solid black lines indicate the median catch per unit effort values, the golden shaded edges show the 95% confidence intervals, and the gray rugs on the upper and lower axes are the positive and negative residuals, respectively.

CPUE values at a depth of 16–35 m (Fig. 4), while in the semi-industrial fleet, we observed somewhat higher CPUE values at 35–70 m, and again at 180–240 m (Fig. 3).

In the temporal component, the CPUE values for both fleets presented similar behavior, with a pronounced drop from 2013 onwards, except for an increase in 2015 (Figs. 3 and 4).

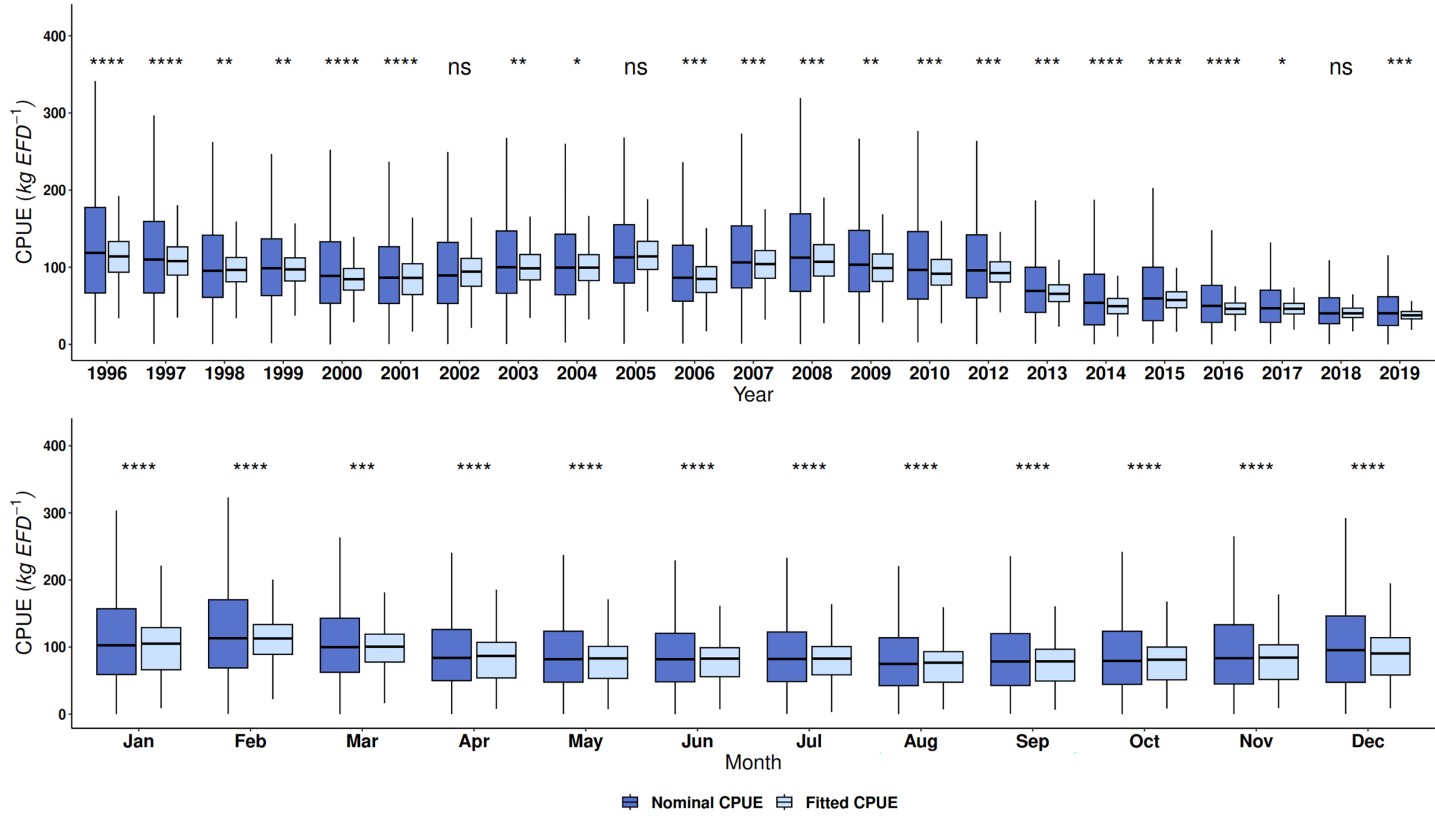

**Figure 5 Nominal catch per unit effort and fitted catch per unit effort by year and month of red grouper recorded by the semi-industrial fleet in the southeastern Gulf of Mexico using the Wilcoxon test.** Symbols indicating significance in the Wilcoxon test: [ns]$p > 0.05$, [*]$p < 0.05$, [**]$p < 0.01$, [***]$p < 0.001$ and [****]$p < 0.0001$.

However, in the semi-industrial fleet, we observed that in previous years, the red grouper CPUE had remained relatively constant (1996–2012; Fig. 3). Regarding the months, we could distinguish two periods: the period with the lowest CPUE values (May–August) and the period with the highest CPUE values (September–March), even though both fleets recorded a higher number of trips during the period with lower CPUE values (Figs. 3 and 4). During the red grouper closure period in the SGM (February–March), the GAMs showed high CPUE values (Figs. 3 and 4). When the closure began in 2003, it was only for one month (from February 15 to March 15). Thus, there was no reduction in the fishing effort because trips were concentrated before and after the closure. It was not until the length of the closure was extended to 2 months in 2017 (from February 1 to March 31) that the number of trips were almost zero.

The spatial component indicated that the CPUE recorded by the semi-industrial fleet decreased from east to west, with the highest CPUE values in the northeastern quadrants (1-B, 5, 6, 7, 11, and 13; Figs. 2 and 3). Besides, most of the fishing effort were concentrated in these quadrants (43.59% of the total trips). However, for the small-scale fleet, we could not identify a clear trend of the influence of place of origin on the CPUE due to the wide uncertainty of the intervals (Fig. 4).

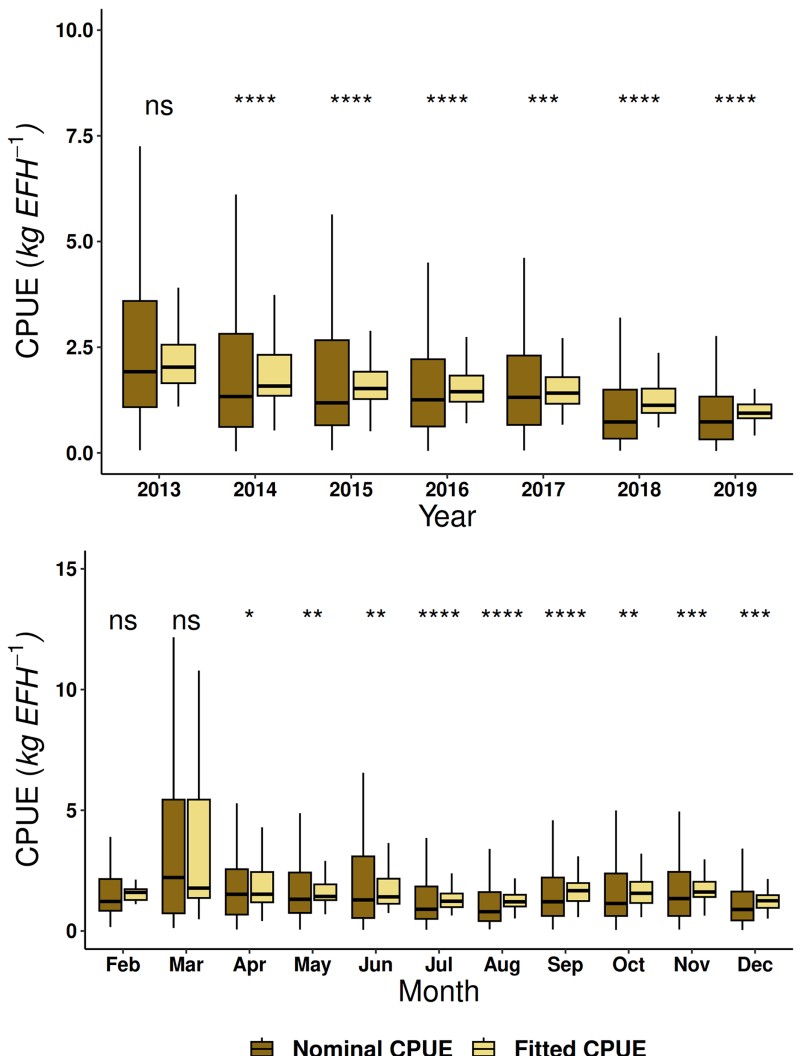

**Figure 6 Nominal catch per unit effort and fitted catch per unit effort by year and month of red grouper recorded by the small-scale fleet in the southeastern Gulf of Mexico using the Wilcoxon test.** Symbols indicating significance in the Wilcoxon test: ns $p > 0.05$, *$p < 0.05$, **$p < 0.01$, ***$p < 0.001$ and ****$p < 0.0001$.

In the operational component, the trend of decreasing CPUE values with the number of navigation days in the semi-industrial fleet was clear for the period between 9 and 25 navigation days (Fig. 3). Nevertheless, we noted wide uncertainty for the rest of the days, which are the ones with the lowest number of fishing trips (<200). It is worth noting that over the years, the semi-industrial fleet has increased its number of navigation days and decreased the number of fishing trips made during the year. This strategy has aimed to improve the efficiency and economic profitability of its fishing activities, because of the low availability of the resource and the increase in variable costs (fuel consumption) of the fishing trips made by this fleet; the higher costs require an increase in income. For both fleets, carrying more crew members led to higher CPUE values (Figs. 3 and 4), possibly due to an increased fishing effort. However, the effect of the change in crew size is unclear due

to the uncertainty of the confidence intervals. Specifically, on some occasions, even if the fishing effort is increased, higher CPUE values are not necessarily guaranteed (in the semi-industrial fleet, it is not common to operate with more than 17 crew members and in the small-scale fleet, with more than four crew members). The fishing gear that had the greatest impact on the red grouper CPUE for both fleets were trips that used longlines and fishing methods with the greatest number of hooks (Figs. 3 and 4). It is remarkable that in the semi-industrial fleet, CFG use began in 2013 and was associated with a decrease in the resource. For the small-scale fleet, SD registered considerable CPUE values.

For both fleets, the GAM proved to be a suitable method to fit the red grouper CPUE because this approach decreased its dispersion (Figs. 5 and 6). The values of the nominal series were higher than the fitted series, although the trend was similar by year and month. Wilcoxon's test showed significant differences for each fleet, except for the years 2002, 2005, and 2018 in the semi-industrial fleet. For the small-scale fleet, there were no differences in the year 2013 and during the months of February and March (Figs. 5 and 6).

## DISCUSSION

Here, we considered several components that could change the distribution and abundance of red grouper in the SGM (*Hernández & Seijo, 2003*; *Monroy-García, Salas & Bello-Pineda, 2010*; *Arreguín-Sánchez et al., 2017*). We have determined, for the first time, the influence of these factors on the red grouper CPUE in the SGM.

The use of reliable and representative indices of abundance is a prerequisite for accurate assessments of fishery resources. This approach allows the establishment of appropriate management strategies and reference points. However, in fisheries—including the red grouper fishery in the SGM, where stock recovery is complex by fleet dynamics and large numbers of users—data quality is very important. For this reason, proportionality is necessary when using the CPUE, which is widely included as an input in dynamic biomass models (*Maunder & Punt, 2004*; *Haddon, 2011*; *Hua et al., 2019*). In conjunction, determining the factors that affect the CPUE will help to counteract the negative trend in catches by identifying strengths and weaknesses in the management strategies employed, besides making it possible to explore and propose complementary actions to strengthen fisheries management.

The influence of the sea surface temperature on red grouper behavior probably suggests changes in thermal preferences during their life stages, which could be linked to their feeding habits and reproduction. In the SGM, opposite spatial movements of red grouper juveniles and adults have been reported (*López-Rocha & Arreguín-Sánchez, 2013*). The movement of juveniles along the coast could be because they seek warm waters that provide food and refuge while avoiding intraspecific competition. *López-Rocha & Arreguín-Sánchez (2013)* stated that red grouper juveniles prefer coral and seagrass substrates, which occur mainly in the warm waters of the coastal systems of central and eastern Yucatan. These areas function as nurseries for several grouper species. In contrast, red grouper adults prefer deeper habitats with sandy bottoms and tend to disperse throughout the SGM, except during their reproductive season (*López-Rocha & Arreguín-Sánchez, 2013*). *Sullivan & Garine-Wichatitsky (1994)* reported that the optimum growth

temperature for red grouper juveniles is between 24 and 26 °C, while warm waters have been associated with inhibiting the reproductive process of adults. Due to climate change, this situation has intensified in recent decades (*Arreguín-Sánchez et al., 2017*, *2019*).

We found that, despite the already reported overlap of the fleets (*Salas, Torres-Irineo & Coronado, 2019*), each fleet is more efficient at the depths where they do not interact with each other. We did not consider that the medium-scale fleet operated at a depth of 40–60 m (Figs. 3 and 4) because the official records classify these trips as a small-scale fleet, despite the fact that the medium-scale fleet has greater autonomy (~3 days) and fishing power, and they generally reported their origin port in Dzilam de Bravo (*Salas, Torres-Irineo & Coronado, 2019*). In this case, the small-scale fleet trips that recorded this depth range have their port of origin in the west (Celestun and Sisal). The second increase in CPUE values recorded by the semi-industrial fleet may be associated with the presence of submarine escarpments—ideal for the benthic habits of larger red grouper adults—on the edge of the Campeche Bank (*Mendoza & Ortiz-Pérez, 2000*).

Year was one of the most important variables to explain the variation in the red grouper CPUE. This finding provides robustness to the index we generated because the objective of CPUE modeling is to generate an accurate annual index that can be used in stock assessments models to support decision-making in fisheries management (*Hinton & Maunder, 2004*; *Maunder & Punt, 2004*). We noted a decrease in the red grouper CPUE during the overexploitation phase, which is consistent with the decrease in catches of this species in the SGM (*Diario Oficial de la Federación (DOF), 2014*). The change in the red grouper CPUE before and after 2012 in the semi-industrial fleet emphasizes that something happened in the fishery and resulted in the decline in the abundance of the resource. We are unable to state with certainty whether this decline was caused by fishing pressure (an increase in a number of hooks, vessels, and fishers), by natural phenomena (historical period of greater activity of hurricanes and tropical storms, 2001–2011; the high impact of red tide, which was most severe in 2002, 2004, 2008, 2009, and 2011), or an interaction of these. However, the use of CFG by the semi-industrial fleet since 2013 is an indicator of fishers' adaptation to this reduction.

The decline in the red grouper CPUE in the middle of the year could be attributed to the availability of the resource and the effort allocated to the red grouper catch. The decrease in effort (fishing trips targeting red grouper) from September onwards is related to fleet dynamics and short-term decisions made by fishers, involving changes in target species and fishing gear to aim at catching octopus (August–December) and lobster (July–February) (*Monroy-García, Salas & Bello-Pineda, 2010*; *Salas, Torres-Irineo & Coronado, 2019*). This is reflected in the decrease in the use of the main gears to catch red grouper (LL, DS, SL, and H), increasing the use of SD and JH, leading to less competition for access to the resource. As indicated by *Hernández & Seijo (2003)*, the octopus season has a positive effect on red grouper because it decreases fishing pressure on the resource, dampening the fishing-related mortality of red grouper in the SGM. This reduced pressure could increase the CPUE and conceal the real abundance of red grouper, as evidenced by the high CPUE recorded in present study during the first months of the year. A natural phenomenon that also reduces the effort from November to February, because the

authorities close the ports to navigation, is the season called *nortes* (*Hernández & Seijo, 2003*); it is characterized by strong northeast winds, precipitation due to polar fronts, and low temperatures (~23 °C). In the context of multi-species and multi-gear fleets, this pattern of effort allocation behavior may be evidence of a change in fishing tactics due to: (i) resource scarcity, (ii) the search for more valuable species, (iii) market demand for a particular species, and (iv) changes in management regimes.

The effective reduction in effort in the red grouper fishery during the closed season was not achieved until its extension to 2 months in 2017. However, in addition to this measure, other factors that have contributed to its compliance are the recognition by fishers of the overexploitation of the resource, as well as the implementation of support programs for fishers ("Temporary Employment Program" and "Respect the Grouper Closed Season"; *Diario Oficial del Gobierno del Estado de Yucatán (DOGEY), 2010*, *2020*), awareness campaigns aimed at fishers and consumers about the regulations ("I take care of the grouper"; *CEDEPESCA, 2018*), and the promotion of various activities that generate economic benefits during the red grouper closure ("Grouper closed season festival"; *Secretaría de Pesca y Acuacultura Sustentables (SEPASY), 2019*). In the Philippines, the implementation of these policies during the closed season has proved to be effective in improving information flow among fishers (*Rola et al., 2018*; *Macusi et al., 2021*), highlighting the key role played by women in participating in awareness campaigns and supporting compliance with the closure (*Macusi et al., 2022*). This has led to greater cooperation between the different sectors and has allowed for a reduction in fishing effort. Moreover, the importance of generating alternative sources of employment for fishers during the closure is emphasized, which increases their income and social benefits (*Rola et al., 2018*; *Macusi et al., 2021*, *2022*). Nevertheless, illegal fishing exists in some regions of the SGM during these months and is associated with a lack of vigilance in the region.

Several authors have reported high red grouper catches in the northeastern quadrants (*Hernández & Seijo, 2003*; *López-Rocha & Arreguín-Sánchez, 2013*) that are linked to their movement during the reproductive season (winter–spring); these movements benefit from the intensified upwelling of Cabo Catoche (*Merino, 1997*). This also explains the high CPUE values recorded by the semi-industrial fleet from November to February.

As suggested by *López-Rocha & Arreguín-Sánchez (2013)*, this area would be a key fishing restriction. In the small-scale fleet, we expected that sites with more developed port infrastructure and technological capabilities of the boats (Progreso or Dzilam de Bravo) would have higher CPUE values, which was not reflected in the results. An example of this contrast is Celestun and Chuburna: Both areas registered high CPUE values, but they have very different fishing contexts and have opted for different fishing strategies. In Celestun, fishers land in the harbor, where there are more boats and fishers; some boats known locally as *lanchones* have been equipped with two outboard motors, each with power between 60 and 115 HP, to make trips of up to 11 days of fishing. Besides these modifications, the *lanchones* use LL with 2,000–5,000 hooks, with similar or greater effort than the vessels of the semi-industrial fleet, and Celestun has intense illegal fishing problems. In Chuburna, on the other hand, fishers land on the beach, and it has one of the lowest numbers of boats and fishers in the SGM and does not present intense illegal fishing

problems (*Monroy-García et al., 2019*). This discrepancy in the origin of the boats coupled with the wide uncertainty of their intervals leads us to explore other alternatives to efficiently manage the red grouper fishery.

Fishing effort has been defined as effective fishing days and effective fishing hours to include fishing gear as an explanatory variable because several authors have considered it to be an indicator of changes in the fishery (*Li et al., 2013*; *Forrestal et al., 2019*; *Salas, Torres-Irineo & Coronado, 2019*). The present study was no exception, as it was variable with the greatest impact on the CPUE values recorded for each fleet. Another factor associated with the decrease in the abundance of red grouper that we did not consider in this work, but that we have observed, is the increase in the number of hooks in the gear used, especially when this number exceeds the established regulatory measures (*Diario Oficial de la Federación (DOF), 2015a*). It is necessary to integrate this factor in future research as a way to obtain a more precise red grouper CPUE. The law that prohibits the methods and techniques of capture in Mexican waters (NOM-064-SAG/PESC/SEMARNAT-2013; *Diario Oficial de la Federación (DOF), 2015b*) prohibits spearguns, while the law that regulates the exploitation of grouper and associated species in Mexico (NOM-065-SAG/PESC-2014; *Diario Oficial de la Federación (DOF), 2015a*) prohibits the capture with *fisga* (a type of harpoon to catch larger fish). These laws can be interpreted as two different things. The prevalence and incidence of SD and speargun use during the lobster season and its impact on the red grouper CPUE encourages the need to update sections of the regulations to eliminate gaps in the laws. The observed trend of the crew size and navigation days with the CPUE may mask the profitability of fishing trips, a factor associated with increased operating costs (*Quijano et al., 2018*; *Salas, Torres-Irineo & Coronado, 2019*) and profit sharing among crew members (*Coronado et al., 2020*).

The high contribution of the fishing gear in modeling the CPUE supports the work of *Monroy-García, Salas & Bello-Pineda (2010)* and *Salas, Torres-Irineo & Coronado (2019)* in proposing the métiers scheme (fleet segmentation that allows the integration of groups of vessels or fishing trips with similar characteristics, such as fishing gear, target species, and fishing area) to manage fishery resources in the SGM. The results of the present study also suggest integrating the crew size when defining the métier codes for future research in the region.

Most assessments of red grouper in the SGM have used the nominal CPUE and have not considered the components of CPUE variation (*Diario Oficial de la Federación (DOF), 2014*). The results of these assessments may bias estimates of parameters (*e.g.*, catchability) and values of interest (*e.g.*, biomass level at maximum sustainable yield). The results of the GAM with both fleets indicated that the fitted CPUE was significantly lower than the nominal CPUE; however, the CPUE values showed a decreasing trend. Similar results have been reported for red grouper in east Florida from fishery-independent surveys (*Christiansen, Winner & Switzer, 2018*). Conversely, for the same species, similarities between nominal and fitted series have been reported in the recreational fisheries (headboat) in Alabama and southwest and northwest Florida (*Rios, 2015*; *Sagarese & Rios, 2018*). These authors point out that the similarities between the series may be due to the way the CPUE is estimated (*e.g.*, using the monthly average); the similar strategies of

fishers operating in nearby sites to catch the target species; and the low amount of annual data, which tends to decrease the variation and does not reflect clear trends in the series (*Tian et al., 2009*; *Wu et al., 2021*). Our findings confirm the influence of several components on the red grouper CPUE, so the differences between the nominal and fitted series may be due to the variability in the strategies of the trips of both fleets (*Salas, Torres-Irineo & Coronado, 2019*) and to the inherent complexity of mixed fisheries (*Tzanatos et al., 2005*), such as the one in question, which catches red grouper in the SGM.

Decreasing catches in fisheries and increasing fishing effort are of global concern. Addressing this issue requires the use of reliable indices of abundance that consider environmental, spatial, temporal, and operational variation in fishery, which will allow assessments to be made and reference points to be defined with greater certainty and thus improve management schemes for sustainable resource utilization. We encourage researchers and decision-makers to adopt this holistic approach to achieve responsible and sustainable management.

## CONCLUSION

We have confirmed the hypothesis that various components influence the red grouper fishery in its overexploitation phase. Using a GAM to fit the red grouper CPUE was adequate because it reduced the variability of the data. We reiterate that the identified components that influence the distribution and abundance of the resource will allow us to adapt and direct feasible strategies to strengthen fisheries management. The results encourage exploring and integrating several variables for future modeling of the red grouper CPUE given the nature of the species (bottom temperature); the dynamics of the fleets (grouping fishing trips into métiers); as well as adding the economic component to evaluate the profitability of fishing trips, weather conditions (dry, rainy, and *nortes* season or wind velocity), oceanographic processes in the region (upwellings), and even natural phenomena (red tide). Several questions have arisen given that this is a sequential fishery: Will there be independent variables for each fleet? Will the indices generated be proportional to abundance? Nevertheless, there has been a debate on how to deal with the behavior of fishers and fishing patterns. One possible solution could be synergistic interaction of the different components involved. In short, we have obtained robust results from modeling the red grouper CPUE. The abundance indices we have generated can be used in future assesments and management work.

## ACKNOWLEDGEMENTS

The authors acknowledge and are grateful to INAPESCA and CRIAP-YUCALPETEN for providing the fishing logs and the interviews analyzed in this research.

### Funding

Iván Oribe-Pérez received a scholarship for the postgraduate study (number 854943) from the Consejo Nacional de Ciencia y Tecnología (CONACYT). The funders had no role in

study design, data collection and analysis, decision to publish, or preparation of the manuscript.

## Grant Disclosures

The following grant information was disclosed by the authors:
Consejo Nacional de Ciencia y Tecnología (CONACYT): 854943.

## Competing Interests

The authors declare that they have no competing interests.

## Author Contributions

- Iván Oribe-Pérez conceived and designed the experiments, performed the experiments, analyzed the data, prepared figures and/or tables, authored or reviewed drafts of the article, and approved the final draft.
- Iván Velázquez-Abunader conceived and designed the experiments, analyzed the data, authored or reviewed drafts of the article, and approved the final draft.
- Carmen Monroy-García conceived and designed the experiments, analyzed the data, authored or reviewed drafts of the article, and approved the final draft.

## Data Availability

The raw measurements are available in the Supplemental Files.

## Supplemental Information

Supplemental information for this article can be found online at http://dx.doi.org/10.7717/peerj.16490#supplemental-information.

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
