# Peer review of "Factors affecting the relative abundance in an overfished stock: red grouper (Epinephelus morio) in the Southeastern Gulf of Mexico"

_PeerJ, doi:10.7717/peerj.16490_

## Round 0.1 · original submission · Major Revisions

Dear authors

Your manuscript needs to be deeply improved. Special attention should be paid to the recommendations given by the reviewers regarding methodology and implementation. Without making the suggested changes, the paper could not be considered for publication.

I hope that you will take all suggestions and improvements into consideration in the next modified version of your manuscript.

·

Basic reporting

I have made comments throughout the manuscript to ensure that clear and unambiguous English is used throughout.

It may be worthwhile for the authors to consider adding in subheadings for the introduction in order to improve the organizational structure.

This may be because I tend to look at things through the ecology/ conservation biology point of view, but I think that the authors should spend a bit more time discussing the grouper fisheries and the ecological importance of grouper in the paragraph before they discuss their specific study species.

The authors provide the raw data, code, and code for figures. The code is easy to follow and makes sense. The authors should consider adding in a "Read Me" file to the dataset which states what abbreviations mean, and zone means. This helps to ensure that the science is reproducible.

There is a clear hypothesis and relevant results.

Experimental design

The model design is well thought out. I am curious about what would happen if other environmental covariates were included in the models. For example, I know some Groupers spawn related to the lunar cycle. I am curious if lunar cycle could influence catch? Also wind? Or primary productivity.

The study is well-justified and its importance in the context of fisheries and stock modeling is well-stated.

I have made some comments in the manuscript (esp. in the methods) where I think more detail is necessary to ensure the results are repeatable. For example, in my opinion, it needs to be made clear where the data sourced from and how the authors acquired it.

Validity of the findings

The results are well interpreted. The results of the models are explained well. This paper does a good job of ensuring that the fisheries significance of the work does not get lost in the modeling related more technical results.

The authors show notably that when fishing fleets are not overlapping with each other, they are more efficient. This is very interesting from an ecological and management point of view. The rationale behind this result is well thought out and explained effectively.

The data is provided. The results are statistically sound. The models were selected and interpreted well, based on my knowledge.

Additional comments

I have made comments throughout the attached manuscript.

Reviewer 2 ·

Excellent Review

This review has been rated excellent by staff (in the top 15% of reviews)
EDITOR COMMENT
It comments clearly and in detail all the sections of the manuscript. In addition, it provides guidance on the possible improvements to be made in each of them and points out the failures found. It points out the shortcomings and weaknesses and gives indications on how to solve them. It also suggests considering other existing studies that help to enrich the work. For all these reasons, it is considered a very good review.

Basic reporting

(1) The article structure is as it should be. The figures and tables are essential and useful.
Figures 3-4: I am not sure whether the figure quality is OK for printing as some of the lines seem too coarse, kindly check it out with the journal guidelines.
Also, in these same figures: I guess on the x’x axis of each panel we can see the rugplot (you should mention it in the legend), but what is this on the top of each panel? (also you should describe it in the legend).
Figures 5-6: Please be reminded that each figure should be able to stand by itself, i.e. without the help of the text of the main part of the manuscript. So it is essential to write in the legend that the comparison and the relevant p-values refer to the Wilcoxon test as mentioned in the Results.

(2) The English language and mostly the writing style could be improved. Some but not all instances are e.g.:
Line 67 (and various other occasions in the Introduction): you often use stock assessment in singular tense, but in my opinion as you are talking generally about assessments the plural would be better, so e.g. here it should be: “…are widely used in stock assessments”
Line 79: machine learnings methods
Line 99: “and senescent” should be changed to” and senescence”
Line 119: change to “…whose the number of fishers, vessels, and catch levels are still unknown.”
Line 260: were almost zero.

(3) The literature references are useful for putting the work into context and sufficient field background has been provided.

(4) The work is self-contained and the results are relevant to the hypotheses.

Experimental design

My main reservation about the publication of the work as it is currently standing regards the fisheries data (the section “Data records” -lines 129-161). In this section, where we would expect to see how the data collection was carried out there is no reference to either the sampling technique (how the fishing vessels were sampled, e.g. randomly, was there perhaps some sort of stratified approach or other?) or to the sampling ratio (how many vessels from the total existing ones or how many trips from the total of trips performed were sampled). This is a serious flaw of the study as the readers cannot be aware of how representative the findings are of the study system and whether the data are unbiased. There is relevant information, but it comes from the beginning of the Results section (lines 213-214) where it is stated that “At least 46,542 semi-industrial fleet trips were analyzed from 1996 to 2019, and 2,775 small-scale fishermen were interviewed from 2013 to 2019.” (but I have no idea why the authors use “at least…” as they should know the exact number of fishing trips).
Also relevant information comes from the Introduction where you give an overview of the fleets examined e.g. a small-scale fleet of ~4200 boats (line 112) and the semi-industrial fleet of 536 vessels. Is there any information about the activity patterns of these fleets and anything other relevant in order to define the sampling population (the number of trips that for the entire fleet constitute the statistical population from which you took your samples)? From the way that you present the data collection in lines 131-139 (logbook entries for semi-industrial, personal interview for monthly catches) these may correspond indeed to a high number of fishing days from the total of a fishing vessel or of fishing months for a small-scale fisher (but please be reminded that even for a single fisher to have a complete knowledge would require 7 years * 12 months = 78 interviews) so the readers that are not familiar with the study system and the study area would still have no idea about the coverage of your sample. Also, importantly, they would not have any idea about how representative that sample is due to the lack of any information on sample selection/sampling design. I am sorry, but this is a very important element and if it were a problem, it could invalidate many of the findings.
As a result, more information is required about: (1) the size of the sample in comparison to the sampled population (ideally you would be able to say that e.g. from a total of 1000 boats operating for 10 years -i.e. 365 days per year = 1000 * 10 * 365 = 3.650.000 we have sampled 46.000 fishing trips, but even if you cannot be this clear assemble the relevant information in one concrete and understandable paragraph), (2) the sampling plan (I am not referring to the technical part e.g. regarding interview, logbook etc. but rather how was the sample selected, e.g. simple random sampling, stratified sampling, multi-stage sampling).

Another important reservation regards the data: How sure can you be that from an interview that is supposed to convey information about fishing of an entire month you have a valid estimate of fishing effort at the scale of effective fishing hour? I would already be happy if the number of days at sea per month is accurate with a ±1 day. Naturally, this is a problem that exists for all data coming from small-scale fisheries, but this is why the scale of sampling is daily (you go sampling for a day and you can record that specific day, not the day and a week or a month before it). Is there some sort of validation that the estimates of effort you are using, especially for the small-scale fleet, are not biased? This would have a serious impact on the dependent variable CPUE of your models. On that same note about fishing effort you do not use number of hooks (but, to give proper credit where it is due, there is a reference to this in the Discussion in lines 381-382). More explanation for this and all relevant choices about the definition of your fishing effort is required in the Methodology part in my opinion.

On the same need for clarity: In lines 199-200, could you please list here specifically, which were the (1) continuous and the (2) categorical variables you have used in the GAMs?

Validity of the findings

The dataset and codes for the analyses have been provided. In general, my reservations about the validity are relevant to the previous section of the experimental design.

The conclusion is relevant and linked to the research.

Additional comments

This is the review of the manuscript “Factors affecting the relative abundance in a senescing fishery: Red grouper (Epinephelus morio) in the southeastern Gulf of Mexico”. The manuscript contains some useful data and could be of some value. I have reservations, mostly about the sampling design and its implementation and the quality and quantity of the data collected. Hence my advice is for a major revision and then re-evaluation whether the article can be publishable. Please see the section “Experimental design” of my review for more details.

Further comments regarding elements of the work that should be taken into account by the authors include the following:

(1) While reading the manuscript I couldn’t help wondering about the use of the terms senescence/senescent regarding the fishery evaluated. I am not convinced about neither the rationale nor the utility of the terminology. The authors refer to a document by FAO in lines 47-48 and use some other references throughout the Introduction, but in my opinion the term is more confusing than helpful (also it is hinting towards biological senescence and thus possibly creating the confusion that maybe the fish targeted are in that stage of their life cycle). In my opinion it would be best to remove it from the manuscript. However, in the case that the authors chose to retain it, it would be essential to include a clear and unambiguous definition of what “senescent fisheries” are. Perhaps the entire description of the transience of fisheries from growth to senescence given in the penultimate paragraph of the Introduction (e.g. see lines 99-109) could be moved earlier in the section (without the specific references to the red grouper historical data).

(2) Discussion: I think there is the need for a final, more general, paragraph in the Discussion section. It does not need to be particularly extensive, but to take a step back and give a more general (less focused on the specific area/stock/fishery) take home message coming from the findings of this study for the international audience of the journal. For a moment, forget about the red grouper and think about what could be useful from your findings and their importance for scientists working on similar questions of similar fisheries elsewhere in the globe.

(3) Some other elements of minor importance:
Line 84: What is SGM? I guess it refers to southeastern Gulf of Mexico, but better to give a definition of the acronym the first time you use it.

Lines 124-128: Please rephrase these two sentences to make them clear. What exactly do you mean by “spatial component”?

Line 131: Perhaps you should briefly explain why there are no data from 2011?

Line 157: Please clarify what is escama

Lines 271-274: Please clarify the sentence “For both fleets, it can be observed that carrying more crew members leads to higher CPUE values (Figs. 3, 4), possibly due to an increased fishing effort; however, carrying more crew members does not guarantee higher CPUE values, which are reflected in the increase of their uncertainty…” as it is perplexing. Perhaps you were referring to effective effort that is not reflected in your measure of CPUE?

Lines 293-294: I think that the use of indices that are representative of abundance is essential in all stock assessments and relevant analyses of fisheries datasets and NOT only on senescent fisheries. Perhaps it would be better to reformulate the sentence stating that this is a general prerequisite, but of course in these fisheries the need for stock rebuilding/recovery makes the data quality even more important.

---

## Round 0.2 · Minor Revisions

Your work has been greatly improved by the review, but it still needs some minor changes as suggested by the reviewers.

·

Basic reporting

The authors fixed the previous grammatical errors in the paper; as such, clear and unambiguous English is now used throughout. Throughout their manuscript they have added references where I previously suggested additional literature was necessary.

The authors have fixed the previously noted structural issues in the manuscript.

Within figure 2, the authors should change "Kilometer" to "Kilometers".

Experimental design

The authors have responded to all of my comments, and have met the necessary standards for publication.

Validity of the findings

The authors have responded to all of my comments, and have provided robust, statistically sound, interpretations of their results.

Reviewer 2 ·

Basic reporting

No comment

Experimental design

No comment

Validity of the findings

No comment

Additional comments

I would like to congratulate the authors for the effort they have invested in the review of the manuscript “Factors affecting the relative abundance in an overfished stock: Red grouper (Epinephelus morio) in the southeastern Gulf of Mexico”. In my opinion, the manuscript is clear and should be accepted for publication. Some minor changes that the authors might want to consider incorporating are listed below.

The aim of the study appearing in lines 87-92 seems a bit misplaced. In my opinion it would be more fitting to place it after the section “Fishery background” and just before the Materials and Methods.

Line 88: for the red grouper stock

Lines 151-152: whose the number of fishers, the vessel sizes and catch levels are unknown.

Line 220: (used during the octopus season)

Line 225: Perhaps the term “monthly compositions” is not the best, but (since I am not an expert in meteorology) the case might be that it is some technical jargon. Kindly check that it is correct in conveying what you are trying to say.

Lines 386-387: ideal for the benthic habits of larger red grouper adults

Line 431: Moreover, the importance

Line 455: Fishing effort has been defined as effective fishing days and effective fishing hours

Lines 463, 465 and 467: if you are referring to legislation here, perhaps it is better to use “law” instead of “rule”.

---

## Round 0.3 · accepted · Accept

I am pleased to confirm that your paper has been accepted for publication in PeerJ.

Thank you for submitting your work to this journal.

With kind regards,

---

## Author Rebuttal · Round 0.3

**Centro de Investigación y de Estudios Avanzados del IPN Unidad Mérida**

**Departamento de Recursos del Mar**

24 October, 2023. Merida Yucatan, Mexico.

**Dear Editorial Board**

**PeerJ.**

We submit the revised version of the paper entitled **Factors affecting the relative abundance in an overfished stock: Red grouper (*Epinephelus morio*) in the southeastern Gulf of Mexico** by Iván Ali Oribe-Pérez, Iván Velázquez-Abunader and Carmen Monroy-García, to be considered for publication in the PeerJ, after completing the review and replying the suggestions made by the reviewers.

We expect the revised version of the manuscript can fulfill the requirements of the journal. We attach the response to specific queries of the reviewers.

We also want to thank the comments and suggestions of two referees, who have helped us improve our manuscript.

Looking forward prompt response, receive our kind considerations

Sincerely,

Iván Velázquez–Abunader

Marine Resources Department

jvelazquez@cinvestav.mx

| Reviewer 1 (Julia Saltzman) | Status | Response |
|---|---|---|
| Basic reporting<br><br>The authors fixed the previous grammatical errors in the paper; as such, clear and unambiguous English is now used throughout. Throughout their manuscript they have added references where I previously suggested additional literature was necessary.<br><br>The authors have fixed the previously noted structural issues in the manuscript.<br><br>Within figure 2, the authors should change "Kilometer" to "Kilometers". | Agree<br><br><br><br>Agree<br><br>Agree | The authors are grateful for the valuable comments provided by the reviewer, which have contributed to improving the paper.<br><br><br>The authors thank the reviewers for their suggestions to improve the structure of the manuscript.<br><br>Within Figure 2, the word "kilometers" was used. |
| Experimental design<br><br>The authors have responded to all of my comments, and have met the necessary standards for publication. | Agree | The authors are grateful for the valuable comments provided by the reviewer. |
| Validity of the findings | | |

| | Status | Response |
|---|---|---|
| The authors have responded to all of my comments, and have provided robust, statistically sound, interpretations of their results. | Agree | The authors are grateful for the valuable comments provided by the reviewer. |
| **Reviewer 2 (Anonymous)** | **Status** | **Response** |
| Basic reporting<br><br>No comment | | |
| Experimental design<br><br>No comment | | |
| Validity of the findings<br><br>No comment | | |
| Additional comments<br><br>I would like to congratulate the authors for the effort they have invested in the review of the manuscript "Factors affecting the relative abundance in an overfished stock: Red grouper (Epinephelus morio) in the southeastern Gulf of Mexico".<br>In my opinion, the manuscript is clear and should be accepted for publication. Some minor changes that the authors might want to | Agree<br><br><br>Agree | The authors are grateful for the valuable comments provided by the reviewers, which have contributed to improving the manuscript.<br><br>The authors are grateful for reviewer's suggestions and are willing to incorporate them into the manuscript. |

| | | |
|---|---|---|
| consider incorporating are listed below | | |
| The aim of the study appearing in lines 87-92 seems a bit misplaced. In my opinion it would be more fitting to place it after the section "Fishery background" and just before the Materials and Methods. | Agree | The authors have placed the aim of the study in the section suggested by the reviewer (see lines 147-152). |
| Line 88: for the red grouper stock | Agree | We have used the suggestion by the reviewer (see lines 147-149). |
| Lines 151-152: whose the number of fishers, the vessel sizes and catch levels are unknown. | Agree | We have used the suggestion by the reviewer (see lines 145-146). |
| Line 220: (used during the octopus season). | Agree | We have used the suggestion by the reviewer (see line 220). |
| Line 225: Perhaps the term "monthly compositions" is not the best, but (since I am not an expert in meteorology) the case might be that it is some technical jargon. Kindly check that it is correct in conveying what you are trying to say. | Revised | We have revised the term and it is more appropriate to use "monthly composites" (see lines 224-226). This expression clearly conveys the idea and is widely used in several research papers:<br><br>Hobson et al. (2008). https://doi.org/10.1016/j.dsr.2007.11.003<br>Martinez et al. (2011). https://doi.org/10.1029/2010JC006836 |

| | | Hu et al. 2023. https://doi.org/10.1016/j.rse.2023.113515 |
|---|---|---|
| Lines 386-387: ideal for the benthic habits of larger red grouper adults. | Agree | We have used the suggestion by the reviewer (see lines 385-387). |
| Line 431: Moreover, the importance | Agree | We have used the suggestion by the reviewer (see lines 431-433). |
| Line 455: Fishing effort has been defined as effective fishing days and effective fishing hours | Agree | We have used the suggestion by the reviewer (see lines 455-458). |
| Lines 463, 465 and 467: if you are referring to legislation here, perhaps it is better to use "law" instead of "rule". | Revised | We have used the suggestion by the reviewer (see lines 463-470). |